# Prediction of Dispersion Rate of Airborne Nanoparticles in a Gas-Liquid Dual-Microchannel Separated by a Porous Membrane: A Numerical Study

**DOI:** 10.3390/mi13122220

**Published:** 2022-12-14

**Authors:** Zohreh Sheidaei, Pooria Akbarzadeh, Carlotta Guiducci, Navid Kashaninejad

**Affiliations:** 1Faculty of Mechanical and Mechatronics Engineering, Shahrood University of Technology, Shahrood 3619995161, Iran; 2Laboratory of Life Sciences Electronics, École Polytechnique Fédérale de Lausanne, 1015 Lausanne, Switzerland; 3Queensland Micro- and Nanotechnology Centre, Nathan Campus, Griffith University, 170 Kessels Road, Brisbane, QLD 4111, Australia

**Keywords:** lung-on-a-chip, porous membrane, gas–liquid dual-channel chip, nanoparticle, numerical simulation

## Abstract

Recently, there has been increasing attention toward inhaled nanoparticles (NPs) to develop inhalation therapies for diseases associated with the pulmonary system and investigate the toxic effects of hazardous environmental particles on human lung health. Taking advantage of microfluidic technology for cell culture applications, lung-on-a-chip devices with great potential in replicating the lung air–blood barrier (ABB) have opened new research insights in preclinical pathology and therapeutic studies associated with aerosol NPs. However, the air interface in such devices has been largely disregarded, leaving a gap in understanding the NPs’ dynamics in lung-on-a-chip devices. Here, we develop a numerical parametric study to provide insights into the dynamic behavior of the airborne NPs in a gas–liquid dual-channel lung-on-a-chip device with a porous membrane separating the channels. We develop a finite element multi-physics model to investigate particle tracing in both air and medium phases to replicate the in vivo conditions. Our model considers the impact of fluid flow and geometrical properties on the distribution, deposition, and translocation of NPs with diameters ranging from 10 nm to 900 nm. Our findings suggest that, compared to the aqueous solution of NPs, the aerosol injection of NPs offers more efficient deposition on the substrate of the air channel and higher translocation to the media channel. Comparative studies against accessible data, as well as an experimental study, verify the accuracy of the present numerical analysis. We propose a strategy to optimize the affecting parameters to control the injection and delivery of aerosol particles into the lung-on-chip device depending on the objectives of biomedical investigations and provide optimized values for some specific cases. Therefore, our study can assist scientists and researchers in complementing their experimental investigation in future preclinical studies on pulmonary pathology associated with inhaled hazardous and toxic environmental particles, as well as therapeutic studies for developing inhalation drug delivery.

## 1. Introduction

As a main component of the lungs, alveoli, with a sizeable capillary interface, enable mass transfer and gas exchange in the human body through their air–blood barrier (ABB) (see Figure 1a) [1]. They are consistently exposed to various inhaled NPs, which due to their small size, can easily transfer throughout the airways and reach them. Some NPs are deposited on the alveolar epithelial cells, while others tend to cross the high-permeable ABB and transfer toward other organs and tissues through the vascular system [2,3,4]. Over the last few decades, morbidity statistics have demonstrated a significant increase in severe pulmonary disorders, including asthma, emphysema, chronic obstructive pulmonary disease, and lung cancer, and highlighted inhaled exposure of alveoli to ambient toxic NPs as the main cause [5,6]. Moreover, new evidence unexpectedly reveals a growing list of other extrapulmonary diseases associated with the translocation of inhaled toxin NPs from the alveoli to the other organs such as brain, kidney, and liver [7]. On the other hand, this transfer mechanism has been recently taken into account in inhalation therapies to develop inhaled nano-drug delivery systems as a targeted and noninvasive method for the treatment, with fewer side effects, of pulmonary and even other organs’ diseases [5]. Therefore, all of this results in a burgeoning demand for pathology, toxicology, and therapeutical studies linked with the inhaled NPs in the alveolar region in order to assess the adverse and beneficial effects of inhaled NPs on the target organ or tissue.

Lung-on-a-chip devices have recently drawn attention in biomedical studies owing to their unique potential in bio-mimicking the physiology of the lung alveolar ABB and providing more accuracy in cellular responses as compared to conventional cell-culture models. First introduced by Huh et al. [8] in 2010, the lung-on-a-chip comprises air/blood (alveolar/vascular) microchannels separated by a thin, porous membrane to enable the co-culturing of epithelial and endothelial cells neighboring on the opposite sides of the membrane and two side vacuum chambers to emulate a breathing lung (see Figure 1b). Adventitiously, this device has opened new research doors in toxico-pharmacological fields, in addition to promising an enormous capacity in recapitulating complex biological pulmonary malfunctions such as edema formation and thrombosis [9,10]. Lung-on-a-chip devices could be conveniently utilized to inject toxic and medical aerosol NPs to replicate inhaled delivery into the alveoli for pathological and therapeutic studies, avoiding unnecessary clinical trials and animal tests. While the air interface in the lung-on-a-chip makes it tremendously unique among the other common liquid-based cellular microsystems, most of the relevant experimental studies aim at streaming an aqueous solution of NPs [8,11,12], and exposure of cultivated cells to the aerosols has been disregarded. Controlled injection of NPs into the lung-on-a-chip will return appropriate sedimentation and distribution in the target area, which is critical for attaining the desired physiochemical efficacy. However, a big research gap exists in the comprehensive understanding of the dynamics of NPs in the lung-on-a-chip device. Accordingly, the distribution and deposition of particles on the endothelial cells in the air channel, as well as their translocation to the media channel, are imperative subjects that should be examined carefully. It is evident that the dynamic behavior of particles in microdevices is determined by many crucial factors, such as the geometry of the device and properties of fluid flow and particles. Nevertheless, optimizing all these parameters is challenging for an enhanced delivery process. Complications associated with fabricating the lung-on-a-chip device and especially the membrane with micro-scale pores, aerosol delivery, and transient monitoring of NPs, which consequently lead to costly and laborious optimization procedures, are the prime suspects responsible for such deficiencies in the field [13,14,15].

Many efforts have been directed toward employing numerical approaches to overcome challenges associated with experimental analyses of various cell-based microfluidic devices to improve their functionality [16,17]. A few recent investigations are devoted to studying the dynamic behavior of solid particles in the lung-on-a-chip. According to their research objectives, these studies follow two main perspectives, namely, the Eulerian approach, which focuses on the concentration of particles considering the problem governed by typical convection–diffusion phenomena, and the Lagrangian approach, which deals with individual particles, tracing the trajectory of each particle separately. For instance, Frost et al. [18] numerically investigated the effect of a porous membrane, one modeled by molecular diffusivity, on the molecular convection–diffusion transport in a bilayer microfluidic device. Their results showed that the molecular concentration at the outlet of the bottom channel increased by membrane porosity augmentation, while it was invertedly affected by the upper channel height.

Although molecule-sized particles are widely considered in many biomedical studies because of their simple diffusivity, particular attention has also been recently paid to particles with diameters ranging between 10 nm to 10 µm due to some other advantages, such as high uptake capacity by cells [19,20]. Because the Eulerian method investigates the average behavior of particles with less computational cost, it is often favored over its more precise counterpart, i.e., the Lagrangian approach. However, the outcome of these two perspectives would not converge for investigating large particles’ dynamics, especially when the total number of released particles is relatively low [21]. Using the Lagrangian approach, Arefi et al. [22] studied the deposition of NPs in an air microchannel under a pulsatile bidirectional flow to simulate breathing patterns. They found that increased airflow rate and breathing frequency raise the deposition rate of particles on the substrate of the channel. In another similar study using finite element simulation, Moghadas et al. [23] reported that airborne delivery of NPs to the cellular region in a microchannel is affected by airflow velocity as well as particle diameter. 

The lung-on-a-chip device, which enables particle tracing in air and media phases, allows for simultaneously investigating the particles’ deposition and translocation. Thus, it is an essential prerequisite for precise toxicological and pharmaceutical studies. Despite its great potential in mimicking the in vivo conditions, the dynamics of NPs in a typical lung-on-a-chip device with a porous membrane and air/blood channels have never been investigated, to the best of our knowledge. To bridge this research gap, the current paper aims to conduct a comprehensive numerical parametric study to provide a clear insight into the role of various hydrodynamical and geometrical properties on the dynamics of airborne NPs in the lung-on-a-chip system. The impact of fluid-flow velocity, membrane porosity, and particle diameter as dominant factors is examined on the dispersion of the particles in both air and media channels as well as their deposition and transfer rates.

This paper is organized as follows: the numerical modeling of the lung-on-a-chip, the governing equations, and the boundary conditions are introduced in Section 2; comparative studies against accessible data and an experimental study are conducted in Section 3 to establish the accuracy of the present numerical study; a case study and parameter sensitivity are discussed in Section 4; and finally, Section 5 provides the conclusions and summarizes the results.

## 2. Governing Equations and Boundary Conditions

In the current work, the dynamics of airborne NPs in the gas/liquid (air/blood) channels of a lung-on-a-chip device separated by a thin, porous membrane are investigated against variations in fluid flow velocity, membrane porosity, and particle diameter. For this purpose, a 2D numerical model is developed using Laminar Flow and Particle Tracing for Fluid Flow modules in COMSOL Multiphysics 5.6 finite element software (see Figure 2) [24] to trace the motion of NPs. For simplicity, the system of governing equations is assumed to be one-way coupled, considering only the effect of the fluid regime on particles and ignoring the reactive effect due to the tiny size of particles. Therefore, first, the time-dependent velocity profile of the fluids is calculated in the channels, and then the obtained solutions are employed to solve the particle tracing. Here, it should be noted that small dimensions and low fluid velocity generally result in laminar fluid flow in microchannels and also allow the airflow to be treated as an incompressible fluid. Moreover, with a Knudsen number smaller than the critical value (*Kn*
≅ 0.0004 < 0.01), the continuum hypothesis and no-slip velocity condition are valid for the airflow field in the channel [25]. The Knudsen number for the airflow is calculated using the hard-sphere collision model as follows [26]:(1)$$Kn=\frac{\lambda }{2h}=\frac{k_{B}}{2\sqrt{2}\pi \sigma^{2}\rho_{a}Rh},$$
where λ is the mean free path and h is the channel height. In this model, $\sigma \;\left(=346\;\mathrm{pm}\right)$ demonstrates the collision diameter of fluid molecules, $R\;\left(≅290\;J.K^{-1}/\mathrm{Kg}\right)$ and $\rho_{a}$ represent the specific gas constant and air density, and $k_{B}$ is the Boltzmann constant. 

In many experimental studies, blood flow is mainly replaced by a medium flow to supply the cells with required nutrition in microfluidic devices, which could be conveniently considered as a Newtonian fluid with similar hydrodynamic properties of water at 37 °C [23]. Consequently, the Newtonian and incompressible air and medium phases are governed by continuity and momentum equations as follows [27]:(2)$$\nabla \cdot u_{i}=0,$$
(3)$$\rho_{i}\left(\frac{\partial u_{i}}{\partial t}+u_{i}\cdot \nabla \;u_{i}\right)=-\nabla p_{i}+\mu_{i}\nabla^{2}u_{i}+\rho_{i}g,$$
where u and p represent the fluid velocity field and pressure, and t is the time, while the subscript $\left(i=a,m\right)$ stands for air and media, respectively.

The Lagrangian approach is used to model the dynamics of NPs under the action of gravity and hydrodynamic forces arising from the motion of the fluid:(4)$$m_{p}\frac{du_{p}^{i}}{dt}=F_{d}^{i}+F_{b}^{i}+m_{p}\left(1-\frac{\rho_{i}}{\rho_{p}}\right)g,$$
where $m_{p}$ and $\rho_{p}$ are the particle mass and density, and $u_{p}^{i}$ indicates the particle velocity in air and media channels, respectively. The right-hand side of Equation (4) is the summation of drag force ($F_{d}^{i}$), the Brownian force ($F_{b}^{i}$), and the particle’s buoyant weight, while the gravity acceleration is considered to be perpendicular to the flow direction. The drag force in Equation (4) is determined in accordance with Stokes law as follows [28]:(5)$$F_{d}^{i}=3\pi \mu_{i}\left(u_{i}-u_{p}^{i}\right)d_{p},$$
where $d_{p}$ is the particle diameter. Also, the Brownian force exerted on particles at each time step taken by the numerical solver could be calculated as [29]:(6)$$F_{b}^{i}=\zeta_{i}\sqrt{6\pi k_{B}\mu_{i}T_{i}d_{p}/∆t},$$
where $T_{i}$ is the fluid temperature, which is considered to be at a constant level of 37 °C, and ∆t represents the time step. In this calculation, $\zeta_{i}$, which indicates the direction of Brownian force, refers to a vector whose components are randomly selected with a Gaussian distribution [29]. Inertial lift force contribution is neglected in Equation (4) since the particle Reynolds number is significantly smaller than 1 ($Re_{P}=\rho_{i}u_{i}d_{p}^{2}/\mu_{i}l$), resulting in viscous forces dominancy [30]. 

Fluid flow boundary conditions in the air and media channels are considered as constant inlet velocity, zero outlet pressure, and no-slip on all solid-fluid interfaces. Furthermore, the top surface of the membrane is employed as the sticky boundary condition for the injected airborne NPs allowing evaluation of particle deposition in the air channel. On the other hand, the pass-through boundary condition is applied to the air-media interfaces to allow NPs to enter the media channel unimpededly. It is worthwhile to note that the numerical simulation is terminated once all the transferred particles flow out of the media channel.

## 3. Model Verification

In this section, comparative studies against accessible data as well as an experimental study are conducted to establish the accuracy of the present numerical study. Accordingly, four limiting-case verifications are considered by setting the membrane’s porosity equal to zero. 

### 3.1. Verification of Numerical Results with Analytical Data

In the absence of solid particles, the first verification study compares the dimensionless axial fluid velocity in the middle of the air channel with analytical data for a 2D steady laminar flow between two fixed parallel plates in Ref. [31]. For this purpose, fluid velocity and vertical distance from the substrate (y) are normalized with respect to the inlet velocity of 0.3 mm/s and the channel height of 100 µm. As depicted in Figure 3a, a fair agreement is observed between the results of the current study and those of Ref. [31].

### 3.2. Verification of Numerical Results with Other Numerical Studies

#### 3.2.1. Verification with a 2D Model

Next, the normalized mean concentration of particles with a diameter of 1 µm along the air channel is compared with that obtained by Saidi et al. [21], who developed a customized numerical code to track 2D dynamics of particles under drag and Brownian forces between two parallel plates. Accordingly, a fully developed inlet flow regime with a mean velocity ($\bar{u}$) of 0.05 m/s and channel height (h) of 1 mm is considered. The concentration of particles is calculated by partitioning the computational space into equal bandwidths along the channel length and then dividing the number of particles in each band to the volume of each bandwidth. Finally, distance from the entrance (x) and particle concentration (C) are nondimensionalized, respectively, with respect to a specific length of 3hPe/16, and initial concentration ($C_{0}=10^{5}/m^{2}$) [21]. $Pe=2\bar{u}h/D$ stands for the Peclet number, where $D\;\left(=2.96\times 10^{-11}{\;m}^{2}/s\right)$ is the diffusion coefficient. The outcome, as shown in Figure 3b, demonstrates a good agreement with Ref. [21]. 

#### 3.2.2. Verification with a 3D Model

Subsequently, the correctness of the developed finite element model is examined to evaluate the deposition of airborne NPs. With the definition of “the ratio between the number of deposited particles on the substrate and the number of total released particles” for the deposition rate (ε), the third verification study compares the deposition rate obtained through the current 2D model with that reported for a 3D model developed by Arefi et al. [22]. All adjustments for boundary conditions are taken into account, according to Ref. [22]. Here, the effect of channel depth is also surveyed, enabling the “Shallow Channel” interface in the 2D model. Figure 3c compares the results obtained for a 2D channel with finite and infinite depth and the 3D model of Ref. [22], where the independence of deposition rate and the channel depth as well as an excellent agreement between these three is perceivable.

### 3.3. Qualitative Verification of the Numerical Results with Experimental Data

Besides verification against the analytical and numerical data in the literature, an experimental study was accomplished to support the results of the present numerical model. The experiment was carried out with the injection of particles into a microchannel under a water flow. Therefore, a numerical simulation was developed in accordance with the experimental parametric values to compare the obtained results from the two studies. Accordingly, a polydimethylsiloxane (PDMS) microchannel with rectangular cross-section was fabricated using a silicone mold (see Figure 4a). The mold was made with maskless ultraviolet (UV) lithography method, as shown in Figure 4b. A silicon wafer coated with a positive photoresist was exposed to direct UV laser to transfer the channel design on it. Subsequently, the wafer was dry etched to engrave the microchannel mold. Then, a 10:1 mixture of PDMS was cast on the mold and the microchannel was sealed with a glass slide to finalize the fabrication procedure (see Figure 4b). A syringe pump was used to inject the particles into the channel under different fluid velocities (see Figure 5 for the experimental setup). In order to have a similar inlet boundary condition to that of the numerical model, the particles were first accumulated at the entrance of the inlet and injected altogether into the channel. Fluid injection was stopped once the particles reached the channel outlet or deposited entirely. 

Magnetic particles of diameter 1 µm and 10 µm were used in the experiments to investigate their sedimentation on the channel substrate under three different water flow velocities. In addition to more deposition efficiency due to their higher inertia and dominant buoyant weight, better control on injection and easier tracing are other main reasons for selecting these particles. The experimental parameter values, including the physical properties of particles, fluid velocities, and the dimension of the microchannel, are listed in Table 1.

Figure 6 qualitatively compares the trend of particle deposition obtained from the numerical and experimental studies under four different cases. The experimental results are actual images showing the particle deposition rate on the bottom layer of the microchannels. These images were obtained from two different parts of the channel substrate, one from the first half of the channel and the other from its second half. 

First, the particles with a diameter of 1 µm were injected into the channel with a height of 100 µm under a fluid velocity of 0.3 mm/s. Due to the dominancy of drag force over the buoyant weight, all the particles were washed out from the channel without any deposition on the substrate (data not shown). Then, the dynamics of particles were examined with a lower fluid velocity of 0.1 mm/s and 0.03 mm/s. Under the fluid velocity of 0.1 mm/s, the particles again streamed quickly toward the microchannel outlet, resulting in a low deposition rate along the channel length (see Figure 6a). The fluid velocity, therefore, was reduced to 0.03 mm/s so that the particles could be uniformly distributed all over the channel substrate and demonstrate a higher deposition rate, Figure 6b. 

There is a mild deposition gradient along the channel width due to the parabolic profile of the fluid velocity in the channel. Moreover, on the substrate next to the channel side walls, the deposition rate is very low as a consequence of the no-slip boundary condition. In the second part of the experiment, particles with a diameter of 10 µm were examined under fluid velocities of 0.1 mm/s and 0.3 mm/s (see Figure 6c,d). The height of the microchannel was increased to 300 µm to allow the particles easily to move inside the channel. The distribution of particles on the substrate of the channel reveals that particles with a diameter of 10 µm under the fluid velocity of 0.1 mm/s provide a concentrated deposition at the channel’s entrance. However, an increase in the fluid velocity to 0.3 mm/s drives the particles quickly forward, resulting in a uniform distribution on the channel substrate. The experimental results for deposition and distribution of the particles on the substrate of the microchannel show a good agreement with the numerical results. 

## 4. Numerical Results

The current numerical parametric study investigates the dynamics of airborne NPs in a gas–liquid dual-channel lung-on-a-chip device with a thin, porous membrane using Equations (2)–(4). Accordingly, the impact of fluid flow velocity, membrane porosity, and particle diameter, as dominant factors, are analyzed on the dispersion of the particles in both air and media channels as well as their deposition and translocation. Numerical results are confined only to some specific cases due to computational limitations, as well as for the sake of brevity. The diameter of solid particles is assumed to be ranging between 10 to 900 nm, according to their high deposition rate reported in the alveolar region [4]. The effect of membrane porosity is examined for pore diameters (d) of 3 and 10 µm, and pore to pore distances (p−p) of 5 and 10 µm. The other numerical values used for simulations are all listed in Table 2.

Before presenting the main results, a brief discussion is presented here to investigate the dynamics of NPs injected with an aqueous solution, which substitutes aerosol injection in many relevant experimental studies [8,11,12]. Accordingly, the pore diameter of 10 µm and pore-to-pore distance of 10 µm are taken into consideration, and the deposition rate and transfer rate are compared in Figure 7 for NPs injected via air and media flow into the upper channel of the device. As previously mentioned, the deposition rate is referred to as the proportion of the total released particles that are deposited on the channel substrate. Similarly, the transfer rate defines the ratio between the number of particles transferred to the media channel and the number of released particles. The results obtained demonstrate a noticeable decrease in deposition/transfer rate when media flow is used for the injection of NPs, regardless of their size. To put it clearly, particles follow the fully-developed parabolic profile of media velocity with a lower chance of settling on the substrate or passing through it (data not shown for brevity). Buoyant weight almost fades for submerged particles with a density of the same order as the fluid, which explains the reduction in deposition/transfer rate, especially for large particles. Furthermore, by increasing the fluid viscosity, the drag force dominates the Brownian force, which plays a key role in the deposition of small NPs. Thus, one can draw the conclusion that injecting through an aqueous solution leads to less efficient particle deposition/translocation, and accordingly, air flow injection should be taken into consideration. 

### 4.1. Deposition of Nanoparticles

This section deals with the sedimentation of airborne NPs in the air channel of the lung-on-a-chip device with a focus on the deposition rate and distribution efficiency. Figure 8 depicts the deposition rate against particle diameter for different membrane porosities and air inflow velocities of 0.3 and 1 mm/s. The results show that the deposition rate declines by increasing the particle diameter up to 200 nm regardless of the inlet velocity. The reason behind this decay is related to the dominancy of Brownian diffusion, which is the main particle deposition mechanism for particles of diameter less than 100 nm according to Stokes–Einstein expression [28,36,37]. On the other hand, as could be deduced from Equations (4)–(6), gravity acceleration becomes more dominant by increasing the particle diameter, while the Brownian effect diminishes. This consequently results in higher deposition rates as the particles enlarge, which can be observed in Figure 8 for diameters larger than 200 nm. A comparison within Figure 8a,b implies an overall decrease in deposition rate by intensifying the inlet flow rate, which makes a larger fraction of NPs leave the device through the air channel outlet. However, both cases follow a similar trend except for particles larger than 700 nm, demonstrating a downtrend of deposition rate for inflow velocity of 0.3 mm/s. By comparing the results of Figure 8a and those of Figure 12a, it is observed that more than 95% of these injected big particles tend to either sediment on the air channel substrate or pass through the perforated membrane with a growing transfer rate against the particle size. Therefore, one can conclude that this unexpected behavior is not linked with blowing NPs off the device but rather with a higher transfer rate into the media channel. Indeed, lower inflow velocity magnitude results in slower axial fluid velocity in the boundary layer above the porous area and prolonged regional residence time, which consequently provide more probability for heavier particles to pass through the membrane. The idea is explicitly illustrated through a series of snapshots in Figure 9, where the trajectory of an initially inert 900 nm particle is compared in the porous region for inflow velocities of 0.3 and 1 mm/s. Such behavior also leads to more dependency of larger NPs’ deposition on membrane porosity, which is supported by the divergence of deposition rate curves corresponding with different porosities in Figure 8.

In addition to the high deposition rate of the injected solid NPs, their better distribution on the channel substrate is another significant factor in toxicological and pharmaceutical studies to attain an effective biochemical impact on cultured cells. Here, the distribution of NPs is mathematically described in terms of the mean value and standard deviation of their position [38].

Figure 10 depicts dimensionless distribution indexes of deposited NPs on the air channel substrate under two air inflow velocities, which are normalized with respect to channel length here. Since membrane porosity has no significant effect on statistical characteristics of particle distribution due to the regular alignment of pores, only a single case with a pore diameter of 10 µm and pore-to-pore distance of 10 µm is considered. Distribution efficiency can simply be assessed against an ideal uniform deposition with normalized mean value of 0.5 and a standard deviation of $1/\sqrt{12}$ (shown with a solid red line in Figure 10) [38].

Under an airflow velocity of 0.3 mm/s, both small and large particles demonstrate low values of the mean location and standard deviation, indicating a concentrated deposition at the entrance of the channel. Increasing the airflow velocity would not alter the results unless for particles larger than 600 nm or smaller than 50 nm, giving rise to a significantly widened distribution range. Although the uniform distribution achieved by higher inflow velocity seems to be desirable, it should be noted that the deposition rate would be unfavorably affected in this case. Therefore, depending on the application, a tradeoff between deposition rate and distribution uniformity should be considered to obtain proper particle size and inflow velocity values. To provide a better insight into the problem, snapshots from the distribution of 900 nm particles are provided in Figure 11.

### 4.2. Translocation of Nanoparticles 

This section deals with the translocation of airborne NPs into the media channel of the lung-on-a-chip device, mainly analyzing transfer rate and distribution efficiency. Figure 12 compares the transfer rate of NPs for different membrane porosities and inflow velocities. A higher translocation efficiency seems achievable by increasing the former and decreasing the latter. The obtained results show that for constant membrane porosity and inlet velocity, the transfer rate declines by increasing the particle diameter up to 200 nm, whereafter it starts to go up. Higher transfer rates arise from Brownian diffusivity for small particles and dominancy of gravity force, as well as longer residence time in the porous regions for large particles. 

Distribution of NPs inside the media channel is also of great importance, especially for targeted particle delivery applications. Since submerged particles with a density close to that of fluid would follow the media velocity profile regardless of inflow velocity, the results are presented only for the inlet velocity of 0.3 mm/s. Accordingly, vertical distribution of transferred NPs are assessed at the outlet of media channel utilizing Equations (4) and (5) with N representing the number of transferred particles and $x_{n}$ showing the vertical position of each particle. Here, the average position and standard deviation are normalized with respect to media channel height, and, similar to the previous section, are then assessed against an ideal uniform distribution (shown with the solid red line in Figure 13). Higher average position and smaller standard deviation for large particles in Figure 13 indicate their tendency to accumulate close to the media channel’s top surface, where the endothelial cells are normally cultured. On the other hand, smaller particles demonstrate a higher penetration power, which is due to their higher vertical velocity magnitude while crossing the membrane as well as their krelatively strong Brownian force in the medium. Moreover, a more uniform distribution is observed for medium-sized particles with a diameter range of 100–300 nm. For a better vision of the impact of size, one can compare the snapshots from the distribution of 10 nm particles in Figure 14 with those obtained from 900 nm particles in Figure 11a. 

In addition to distribution efficiency, the current study investigates the transient evolution of transferred NPs as a significant factor in toxico-pharmacokinetic analyses [39]. First, the relative concentration, which is defined by the ratio of total injected particles that flow out through the outlet of the media channel, is recorded, and then, a numerical differentiation is utilized to obtain the time derivative of the data. After a Gaussian curve fitting, final results are depicted in Figure 15 for different membrane porosities and inlet velocities. Figure 15, which is known as the so-called concentration-time plot [38], provides important details describing the behavior of transferred NPs to the media channel, such as area under each curve, which is equal to the transfer rate (see Figure 12), maximum relative concentration rate ($c_{max}$), and time to reach this maximal value ($t_{max}$). For example, it is shown that 10 nm particles demonstrate the highest $c_{max}$, which is significantly augmented by increasing the inflow velocity. Moreover, it is evident that smaller particles display a faster transmission with a lower $t_{max}$. Although the overall value of $c_{max}$ grows by increasing the membrane porosity and inflow velocity, it should be noted that this may also considerably change $t_{max}$ depending on the particle size. 

## 5. Conclusions

With a promising potential in recapitulating in vivo conditions, lung-on-a-chip technology has recently attracted the attention of researchers. This dual-channel device with a sandwiched porous membrane enables particle tracing in air and media phases simultaneously, which is an essential prerequisite for further toxicological and pharmaceutical studies. Despite this remarkable capability, a detailed investigation into the dynamics of NPs in this device has been overlooked by the literature. Here, a comprehensive numerical parametric study was conducted to study the role of various hydrodynamical and geometrical properties on the dynamics of airborne NPs. Accordingly, the impacts of fluid flow velocity, membrane porosity, and particle diameter were examined on the dispersion of the particles in both air and media channels, as well as their deposition and translocation. Consistent with the experimental observation reported in the literature, we numerically showed that the aerosol injection of NPs provided far more efficient deposition/translocation than the aqueous solution. Although very small particles ($d_{p}<50\;\mathrm{nm}$) as well as very large ones ($d_{p}>700\;\mathrm{nm}$) demonstrated a high deposition rate, they tended to sediment close to the entrance of the channel for low inflow velocities, which could be undesirable for many applications demanding a uniform distribution. On the other hand, a higher inflow velocity provided a more uniform distribution; however, it also resulted in lower deposition and translocation rates. Therefore, it can be concluded that medium-sized particles ($500\;\mathrm{nm}<d_{p}<700\;\mathrm{nm}$) are more appropriate for uniform distribution with lower inflow velocities. It was demonstrated that the translocation of NPs into the media channel became more efficient by reducing the particle size as a factor of Brownian diffusivity and also by increasing the particle residence time in the porous region as a consequence of the gravity force’s dominancy in large particles. However, smaller particles showed a better distribution in the media channel than did larger ones. Therefore, the results of this paper propose optimizing particle size as well as inflow velocity and membrane porosity, depending on the objectives of biomedical investigations. Even though the current numerical analysis is carried out with some simplifications, such as neglecting the cultured cell layers, it takes the first step towards a perceptive insight into the dynamics of the NPs in the lung-on-a-chip device. As a reliable complementary method seeking to reduce experimental cost, time and efforts, this study can be beneficial for future preclinical studies on pulmonary pathology associated with inhaled hazardous and toxic environmental particles, as well as therapeutic studies for developing inhalation drug delivery.

## Figures and Tables

**Figure 1 micromachines-13-02220-f001:**
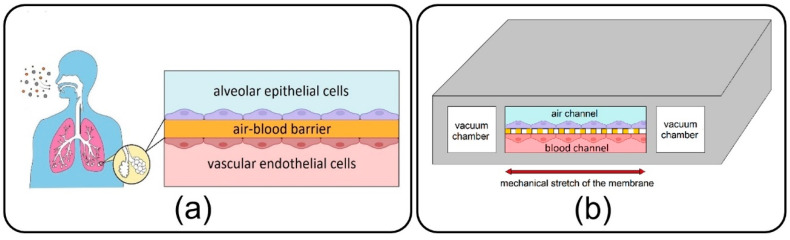
(**a**) Air–blood barrier (ABB) in the alveolar space of human lungs; (**b**) Microfluidic lung-on-a-chip device comprises air/blood channels separated by a thin, porous membrane and two side vacuum channels.

**Figure 2 micromachines-13-02220-f002:**
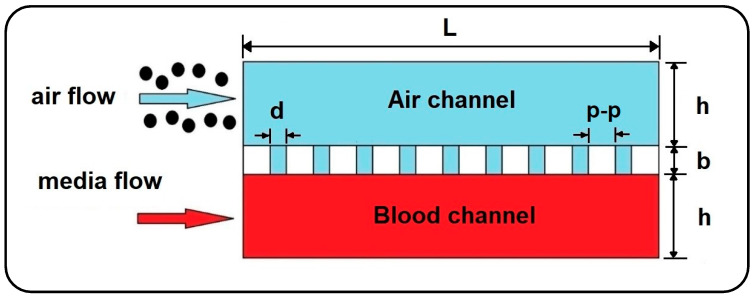
2D model of airborne delivery of NPs into a gas-liquid dual-channel microdevice with a thin, porous membrane.

**Figure 3 micromachines-13-02220-f003:**
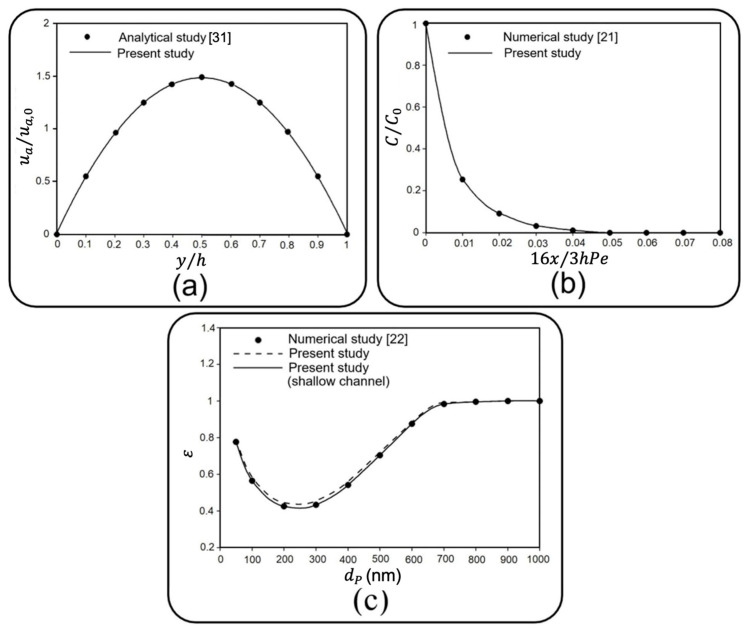
Comparison of the present numerical results with (**a**) analytical data in Ref. [31] for a 2D airflow between two parallel plates. Axial fluid velocity and vertical distance from the substrate are normalized with respect to the inlet velocity of 0.3 mm/s and the channel height of 100 µm. Subsequently, (**b**) numerical data in Ref. [21] is shown for the mean concentration of particles with a diameter of 1 μm along two parallel plates. Distance from the entrance and particle concentration are nondimensionalized, respectively, with respect to a specific length of 3hPe/16, and initial concentration ($C_{0}=10^{5}/m^{2}$). $Pe=2\bar{u}h/D$ is the Peclet number, and where $D\;\left(=2.96\times 10^{-11\;}m^{2}/s\right)$ is the diffusion coefficient. Finally, the (**c**) the numerical result of Ref. [22] for deposition rate of NPs with various diameters on the substrate of an air channel is shown.

**Figure 4 micromachines-13-02220-f004:**
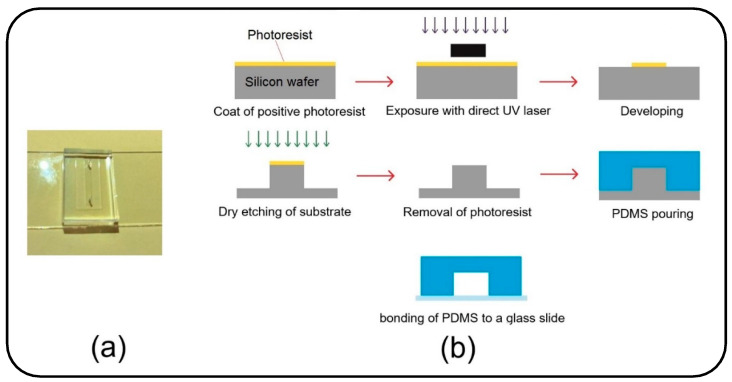
(**a**) Microfluidic device, which is bonded to a glass slide and contains a channel with one inlet and one outlet; (**b**) Schematic diagram of the fabrication process of the microfluidic device.

**Figure 5 micromachines-13-02220-f005:**
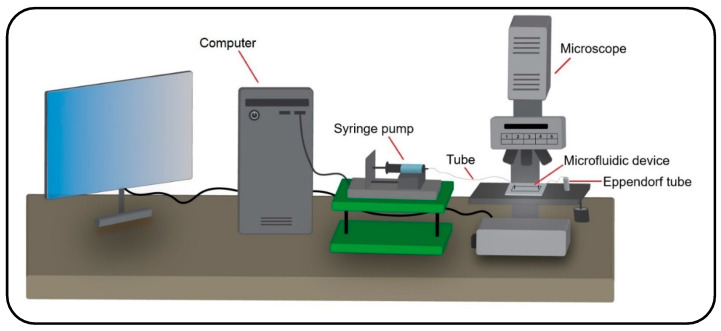
Schematic diagram of the experimental setup for injecting the particles into a microfluidic device using a syringe pump and monitoring their dynamics throughout the device with a microscope.

**Figure 6 micromachines-13-02220-f006:**
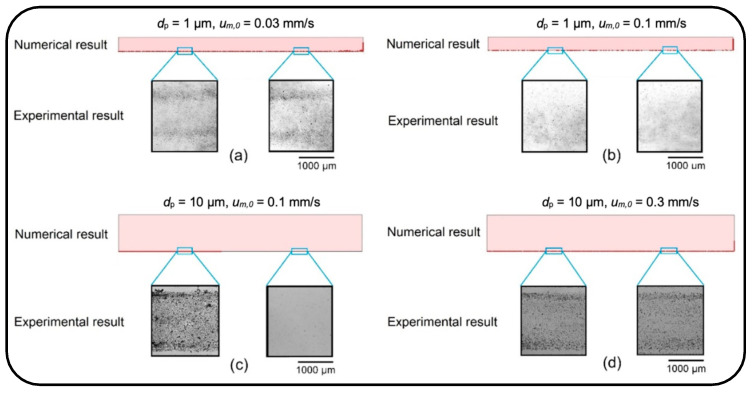
Qualitative comparison of distribution and deposition of the particles on the microchannel substrate obtained from numerical and experimental studies. (**a**) $d_{p}=1$ µm, $u_{m,0}=0.03$ mm/s; (**b**) $d_{p}=1$ µm, $u_{m,0}=0.1$ mm/s; (**c**) $d_{p}=10$ µm, $u_{m,0}=0.1$ mm/s; and (**d**) $d_{p}=10$ µm, $u_{m,0}=0.3$ mm/s.

**Figure 7 micromachines-13-02220-f007:**
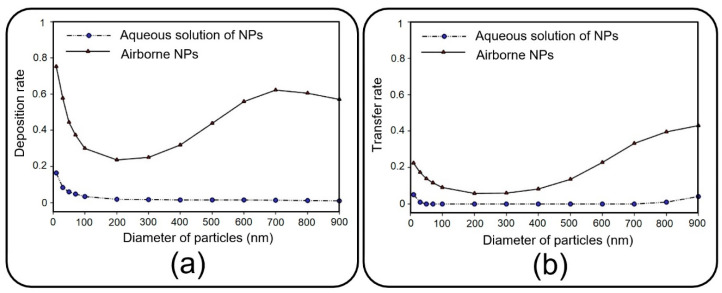
(**a**) Deposition rate and (**b**) transfer rate of the media- and air-induced NPs in the top channel of the lung-on-a-chip device when the fluid velocity is 0.3 mm/s, and both pore diameter and pore-to-pore distance are 10 µm.

**Figure 8 micromachines-13-02220-f008:**
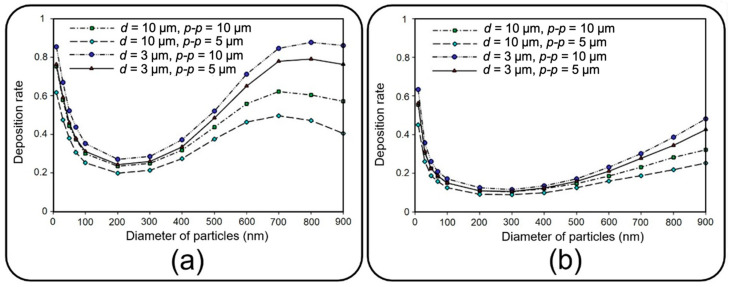
Deposition rate of airborne NPs of diameter ranging from 10 nm to 900 nm on the substrate of the air channel with different membrane porosities and airflow velocities of (**a**) 0.3 mm/s and (**b**) 1 mm/s.

**Figure 9 micromachines-13-02220-f009:**
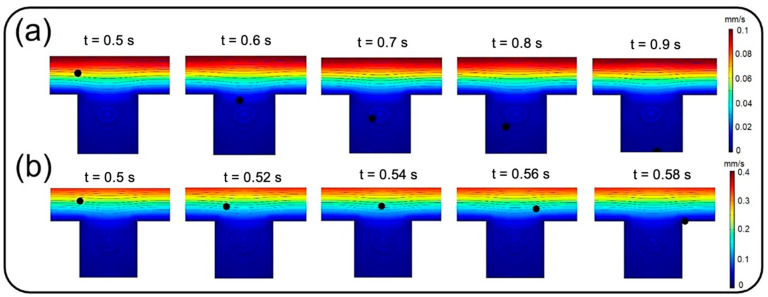
Trajectory of an initially inert 900 nm particle in the porous region for inflow velocities of (**a**) 0.3 mm/s and (**b**) 1 mm/s.

**Figure 10 micromachines-13-02220-f010:**
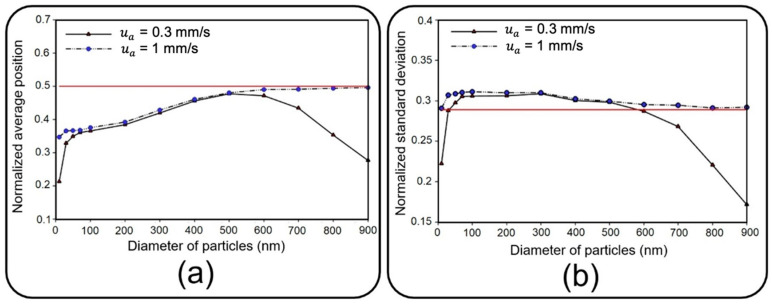
Normalized (**a**) average position and (**b**) standard deviation of deposited NPs of diameter ranging from 10 nm to 900 on the substrate of the air channel with pore diameter and pore to pore distance of 10 µm under two different fluid velocities of 0.3 mm/s and 1 mm/s. The solid red line corresponds with an ideal deposition.

**Figure 11 micromachines-13-02220-f011:**
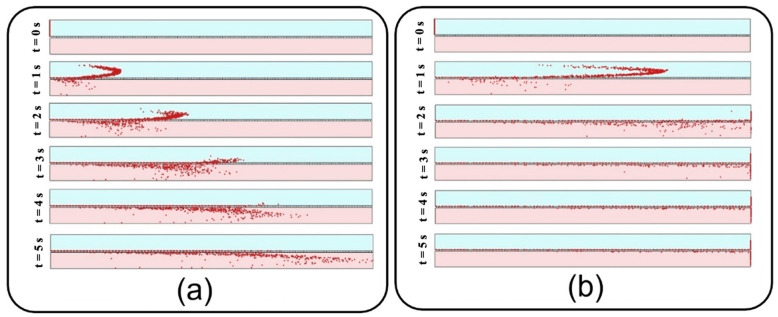
Snapshots from transient distribution of NPs with a diameter of 900 nm in the lung-on-a-chip device with pore diameter and pore-to-pore distance of 10 µm under the fluid velocity of (**a**) 0.3 mm/s, and (**b**) 1 mm/s.

**Figure 12 micromachines-13-02220-f012:**
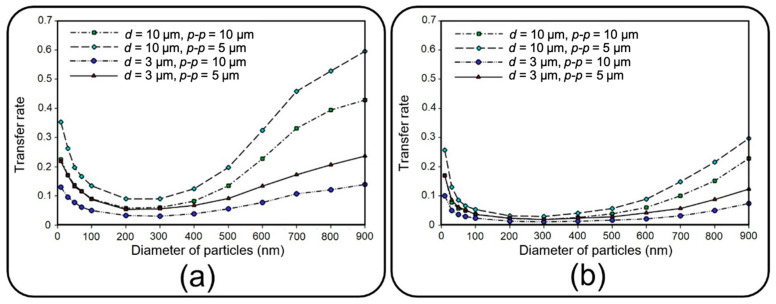
Transfer rate of NPs of diameter ranging from 10 nm to 900 to the media channel of the lung-on-a-chip device with different membrane porosities considering equal inflow velocities of (**a**) 0.3 mm/s and (**b**) 1 mm/s for air and media channels.

**Figure 13 micromachines-13-02220-f013:**
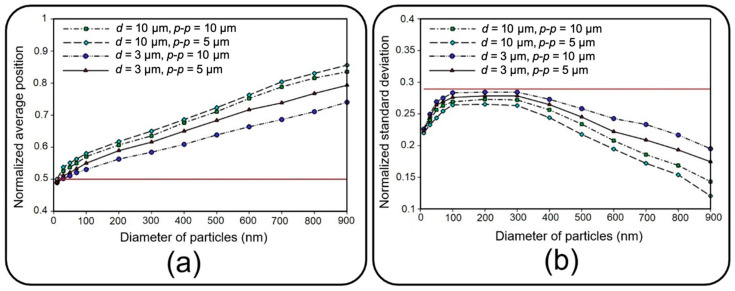
Normalized (**a**) average position and (**b**) standard deviation of transferred NPs of diameters ranging from 10 nm to 900 at the outlet of the media channel in the lung-on-a-chip device with different membrane porosities under the fluid velocity of 0.3 mm/s. The solid red line corresponds with an ideal distribution in the media channel.

**Figure 14 micromachines-13-02220-f014:**
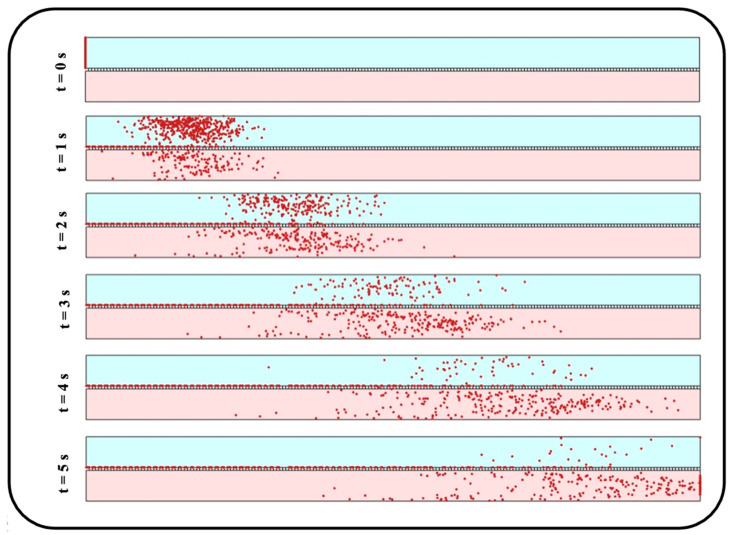
Snapshots from transient distribution of particles with a diameter of 10 nm in the lung-on-a-chip device with pore diameter and pore-to-pore distance of 10 µm and fluid velocity of 0.3 mm/s.

**Figure 15 micromachines-13-02220-f015:**
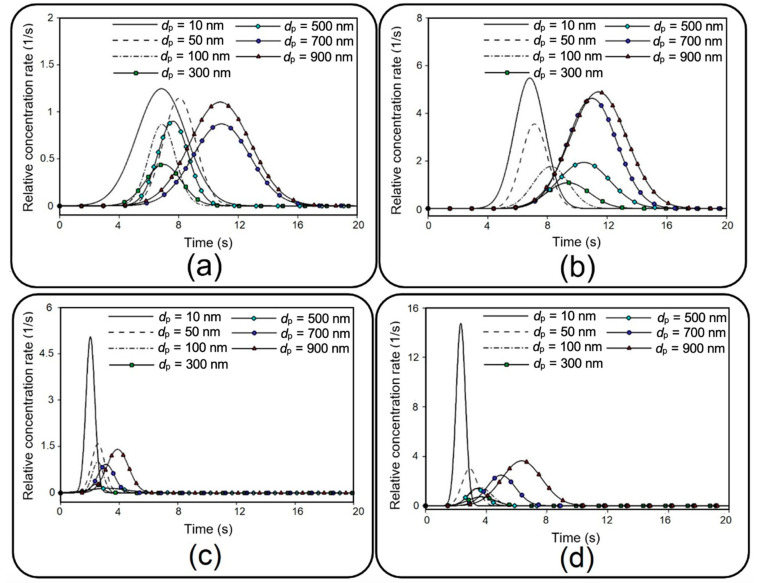
Transient relative concentration rate for NPs with different diameters. (**a**) d=3 µm, p−p=10 µm; $u_{a}=0.3$ mm/s; (**b**) d=10 µm, p−p=5 µm, $u_{a}=0.3$ mm/s; (**c**) d=3 µm, p−p=10 µm, $u_{a}=1$ mm/s; and (**d**) d=10 µm, p−p=5 µm, $u_{a}=1$ mm/s.

**Table 1 micromachines-13-02220-t001:** The values of the parameter used for the experimental test.

Diameter of Particles $\left(d_{p})\;[µm\right]$	Density of Particles $\left(\rho_{p})\;[\mathrm{kg}/m^{3}\right]$	Dimension of Channel $\left(w\times h\times l)\;[µm^{3}\right]$	Fluid Velocity $\left(u_{m,0})\;[\mathrm{mm}/s\right]$	Flow Rate [µL/min]
1	2200	2000 × 100 × 15,000	0.03 and 0.1	0.36 and 1.2
10	1470	2000 × 300 × 15,000	0.1 and 0.3	3.6 and 10.8

**Table 2 micromachines-13-02220-t002:** The values of the parameter used for the numerical simulations.

Parameters	Values	Descriptions	Reference
h	100 µm	Height of the channels	[32]
l	2 mm	Length of the channels	[22]
d	10 µm, 3 µm	Diameter of the membrane pore	[32,33]
*b*	10 µm	Thickness of the membrane	[8]
$\rho_{p}$	1180 kg/m^3^	Density of particles	[22]
$\rho_{m}$	1000 kg/m^3^	Density of media	[34]
$\mu_{m}$	0.718 mPa.s	Viscosity of media	[34]
$\rho_{a}$	1.123 kg/m^3^	Density of air	[35]
$\mu_{a}$	0.019 mPa.s	Viscosity of air	[35]
$u_{m,0}$	0.3 mm/s, 1 mm/s	Media velocity at inlet	[23,32]
$u_{a,0}$	0.3 mm/s, 1 mm/s	Air velocity at inlet	[22]

## Data Availability

The data presented in this study are available on request from the corresponding author. The data are not publicly available due to privacy.

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
