# Peer review of "Prediction of Dispersion Rate of Airborne Nanoparticles in a Gas-Liquid Dual-Microchannel Separated by a Porous Membrane: A Numerical Study"

_micromachines, 2022, doi:10.3390/mi13122220_

Round 1
Reviewer 1 Report
In this work, the authors introduce a model to simulate the dynamic of nanoparticles in a lung on a chip device. The authors provide an extent introduction on the topic providing relevant references and finally, they introduce their model and simulations. I consider this work has potential for being published at Micromachines, however, I consider some relevant details in how the simulations were performed are needed. Therefore, I consider the decision on publishing this work must be reconsider after major revision of the article to make clear the following questions.
- Lines 141, 142: How do you perform the simulations? I mean, do you first calculate a steady flow profile and then you use this solution to apply the particle tracing? Or, do you calculate a time-dependent flow profile at the same time you perform the particle tracing? Do you have considered the presence of the particles perturbs the flow profile or do you neglect this effect? Do you use Equation 3 to solve the particle tracing? Please, specify this in the text.
- Equation 2: In line 174 you say that Re<<1, so, why do you use this version of the Navier-Stokes equation? In a steady laminar flow regime, you can neglect the time dependent term and the uDelta(u) term, improving the convergence of your simulations, why didn’t you use the simplified version for laminar flow?
- Lines 173 and 174: you neglect the lift force based in the Reynolds number, this low Reynolds number just tell you that the viscous forces are bigger than other forces exerted by the fluid in your particle, however, you have other forces independent of the fluid velocity (weigh and Brownian motion). Is this lift force also small when compared with these non-hydrodynamic forces?
- Lines 188 to 194: I don’t understand well what do you do here, Ref [30] is a fluid dynamics text book, are these data taken from any real experiment or are they just the analytical solution of the laminar flow profile among two parallel plates? Isn’t it better compare these data to the analytical flow profile (u=u0*[1-y/h], being u0 the velocity in the center on the channel, y the vertical component and h the height of the channel? Moreover, if you reduce to zero the porosity of the channel (obtaining, consequently the classical case of a laminar flow between two parallel plates), it is generally expected to obtain this flow profile, it is really necessary including this data in the main text? Please, consider including this data in a supporting material (section or similar).
- Line 194: Please, specify how do you measure the concentration.
- Line 197: Some lines before, you use other dimensions for your channel, why do they change now?
- Figure 3: Why do your simulations look like lines connecting the points in the previous studies? Do you have exactly the same points at the same position that these previous works? If not, please, consider including more points of your simulations in these graphs, so the curve they describe could be better appreciate it.
- Figure 4: How do you compare the nanoparticles in aqueous solution and the airborne ones? Do you do the simulation in the same channel but in one case with air and in the other with an aqueous medium?
- Equations 6 and 7: The concepts of mean and standard deviation are widely known and can be easily found at any basic statistics book. Please consider not include them here for the sake of clarity.
- Figure 7b: In this graph, it cannot be appreciated if from 100 nm to 500 nm is there any difference for the 0.3 mm/s and de 1 mm/s curve. Please, consider rearrange the vertical axis to make it easier to see.
- Line 377: the text seems to be incomplete, are there any missing lines in the text?
- Figure 12: In these graphs for some curves you include line and points, while in other curves (in the same plot) doesn’t include the points. Why this happens? Has it any meaning?
Reviewer 2 Report
Review of “Dispersion of Airborne Nanoparticles in a Gas-Liquid Dual-
Microchannel Separated by a Porous Membrane: A Numerical Study”
By Zohreh Sheidaei, Pooria Akbarzadeh, Carlotta Guiducci and Navid Kashaninejad
Submitted to: Micromachines
In this paper, the authors used the software COMSOL to conduct a 2D, laminar, one-way-coupling numerical simulation of NPs dynamics in gas/liquid channels of a lung-on-a-chip device. The parameters investigated are velocity, membrane porosity, and particle diameter. The main concern is that the present results were validated against computational results using more or less the same numerical model. The present model needs to be verified against experimental results to be credible. The authors are encouraged to resubmit their work after such validation is provided. The following points may also help the authors to improve the quality of their manuscript.
Why was laminar flow put between quotations on page 4, line 141?
What is the pressure drop of air through the channel? What is the corresponding change in the air density between the inlet and outlet of the channel? Does this agree with the assumption of compressible flow?
Please provide details on how calculated the Kn number justified the no-slip boundary condition. What was the value of the mean free path used in calculating the Kn number?
Is it unclear to this reviewer how the present results plotted in Fig. 3(a,b,c) are not curves? Instead, they are connected in straight lines. Also, can you comment on how the agreement is 100% with the data from Refs [30, 21, 22]?
Fig. 3 will be more useful for the reader when the actual variables used are plotted on the axes instead of “normalized vertical distance”, and “normalized concentration).
Page 5, line 200: It is unclear what values are used to calculate the Peclet number.
The resolution (dpi) of the graphs is not adequate. Please improve it.
The font in Fig. 6 is too small. Please increase it.
The definitions of the mean and the standard deviation (Equations 6 & 7) are known for every undergraduate student. Please remove it.
Round 2
Reviewer 1 Report
After receiving the answers by the authors I consider this article should be accepted in the present form
Author Response
Thank you so much for your constructive comments. We are so happy that the reviewer has accepted our manuscript in the present form.
Reviewer 2 Report
1. It is unclear how the comparison between the experimental results and the computational results was quantified.
2. The pressure and corresponding density at the inlet and the outlet should be specified.
3. The Kn number calculation should be included in this manuscript.
4. The present results plotted in Fig. 3(a,b,c) should be fitted with the best fit curve instead of being connected with straight segments.
5. Fig. 3 is still not clear. The actual variables should be plotted on the axes instead of “normalized vertical distance”, and “normalized concentration).
Round 3
Reviewer 2 Report
I. Without calculating the Knudsen number and showing to the reader that the flow regime can be approximated by the continuum assumption, the current numerical model may not be correct. This is especially important in the absence of experimental data that can be quantitatively used to validate the current numerical methods. This reviewer is still having concerns regarding the assumption of continuum flow based on the calculated Knudsen number, as follows:
I.1. What are the actual values used to calculate the Knudsen number and what is the calculated value of the Knudsen number? The authors have to report those numbers, d_p, l, and Kn.
I.2. Why is the particle diameter used to calculate the mean free path instead of the scattering cross-sectional area?
I.3. Why is the length used instead of the channel height?
II. The clarity of Fig. 3 should be improved by including the normalized variables on the axis. The authors misunderstood the comment on the last version. What is needed is to replace the words "normalized vertical distance" and "normalized axial flow" by the variables used, e.g. [y/h].
Round 4
Reviewer 2 Report
Thank you for detailing the Kn number calculations.
The scattering cross-section area is not the channel cross-section area. Instead, it represents the collision cross-section area of the molecules. This scattering cross-section area is the molecule cross-section area only for the hard-sphere model. The authors are strongly encouraged to clarify that the hard-sphere model was used in calculating the Kn number.
